# Activation of plant immunity through conversion of a helper NLR homodimer into a resistosome

**Muniyandi Selvaraj**[1], **AmirAli Toghani**[1], **Hsuan Pai**[1], **Yu Sugihara**[1], **Jiorgos Kourelis**[1], **Enoch Lok Him Yuen**[2], **Tarhan Ibrahim**[2], **He Zhao**[1], **Rongrong Xie**[1¤a], **Abbas Maqbool**[1¤b], **Juan Carlos De la Concepcion**[3¤c], **Mark J. Banfield**[3], **Lida Derevnina**[1¤d], **Benjamin Petre**[1¤e], **David M. Lawson**[3], **Tolga O. Bozkurt**[2], **Chih-Hang Wu**[1¤f], **Sophien Kamoun**[1] *, **Mauricio P. Contreras**[1] *

1 The Sainsbury Laboratory, University of East Anglia; Norwich Research Park, Norwich, United Kingdom, 2 Imperial College London, London, United Kingdom, 3 Department of Biochemistry and Metabolism, John Innes Centre, Norwich, United Kingdom

¤a Current address: Joint Center for Single Cell Biology, School of Agriculture and Biology, Shanghai Jiao Tong University, Shanghai, China
¤b Current address: Department of Biochemistry and Metabolism, John Innes Centre, Norwich, United Kingdom
¤c Current address: Gregor Mendel Institute, Austrian Academy of Sciences, Vienna BioCenter, Vienna, Austria
¤d Current address: Crop Science Centre, Department of Plant Sciences, University of Cambridge, Cambridge, United Kingdom
¤e Current address: Université de Lorraine, INRAE, IAM, Nancy, France
¤f Current address: Institute of Plant and Microbial Biology, Academia Sinica, Nankang, Taipei, Taiwan
* sophien.kamoun@tsl.ac.uk (SK); mauricio.contreras@tsl.ac.uk (MPC)

**Data Availability Statement:** All data are available in the main text or the supplementary materials. Data related to the NRC2 structure can be found at PDB/EMDB, with PDB ID 8RFH and EMD-19121,

## Abstract

Nucleotide-binding domain and leucine-rich repeat (NLR) proteins can engage in complex interactions to detect pathogens and execute a robust immune response via downstream helper NLRs. However, the biochemical mechanisms of helper NLR activation by upstream sensor NLRs remain poorly understood. Here, we show that the coiled-coil helper NLR NRC2 from *Nicotiana benthamiana* accumulates in vivo as a homodimer that converts into a higher-order oligomer upon activation by its upstream virus disease resistance protein Rx. The cryo-EM structure of NbNRC2 in its resting state revealed intermolecular interactions that mediate homodimer formation and contribute to immune receptor autoinhibition. These dimerization interfaces have diverged between paralogous NRC proteins to insulate critical network nodes and enable redundant immune pathways, possibly to minimise undesired cross-activation and evade pathogen suppression of immunity. Our results expand the molecular mechanisms of NLR activation pointing to transition from homodimers to higher-order oligomeric resistosomes.

as well as S1 Table. Datasets and scripts associated to NRC phylogenomics analyses can be accessed on Zenodo (https://zenodo.org/records/10354350 and https://zenodo.org/records/13362063).

**Funding:** The authors received funding from The Gatsby Charitable Foundation (MS, AT, HP, YS, JK, HZ, RX, AM, BP, LD, DML, CHW, SK, MPC); The John Innes Foundation (JCDLC); Biotechnology and Biological Sciences Research Council (BBSRC) BB/P012574 (Plant Health ISP) (MS, AT, HP, YS, JK, HZ, RX, AM, BP, LD, DML, CHW, SK, MPC); BBSRC BBS/E/J/000PR9795 (Plant Health ISP – Recognition) (MS, AT, HP, YS, JK, RX, AM, BP, LD, DML, CHW, SK, MPC); BBSRC BBS/E/J/000PR9796 (Plant Health ISP – Response) (MS, AT, HP, YS, JK, RX, AM, BP, LD, DML, CHW, SK, MPC); BBSRC BBS/E/J/000PR9797 (Plant Health ISP – Susceptibility) (MS, AT, HP, YS, JK, RX, AM, BP, LD, DML, CHW, SK, MPC); BBSRC BBS/E/J/000PR9798 (Plant Health ISP – Evolution) (MS, AT, HP, YS, JK, RX, AM, BP, LD, DML, CHW, SK, MPC); BBSRC BB/V002937/1 (LD, SK, MPC); BBSRC BB/T006102/1 (TI, TOB); BBSRC impact acceleration award BB/X511055/1 (ELHY); Academia Sinica and National Science and Technology Council NSTC-111-2628-B-001-023 (CHW); European Research Council (ERC) 743165 (SK). The funders had no role in the study design, data collection and analysis, decision to publish, or preparation of the manuscript.

**Competing interests:** TOB and SK receive funding from industry on NLR biology and cofounded a start-up company (Resurrect Bio Ltd.) on resurrecting disease resistance. JK, LD, SK and MPC have filed patents on NLR biology. LD and MPC have received fees from Resurrect Bio Ltd.

**Abbreviations:** BN-PAGE, blue native polyacrylamide gel electrophoresis; co-IP, co-immunoprecipitation; NB, nucleotide binding; NLR, nucleotide-binding and leucine-rich repeat; STAND, signal transduction ATPases with numerous domains.

# Introduction

The nucleotide-binding and leucine-rich repeat (NLR) class of intracellular immune receptors is an important component of innate immunity across the tree of life. They mediate intracellular recognition of pathogens and subsequently initiate an array of immune responses in order to counteract infection [1,2]. Plant NLRs can be activated by pathogen-secreted virulence proteins, termed effectors, which pathogens deliver into host cells to modulate host physiology [3]. NLRs belong to the signal transduction ATPases with numerous domains (STAND) superfamily. They typically exhibit a tripartite domain architecture consisting of an N-terminal signalling domain, a central nucleotide-binding domain and C-terminal superstructure forming repeats [4]. The central domain, termed NB-ARC (nucleotide-binding adaptor shared by APAF-1, plant R proteins, and CED-4) in plant NLRs, is a signature feature of this protein family and plays a key role as a molecular switch, mediating conformational changes required for activation [4]. A hallmark of NLR activation in eukaryotes and prokaryotes is their oligomerization into higher-order immune complexes, such as plant resistosomes or mammalian and bacterial inflammasomes. These complexes initiate immune signalling via diverse mechanisms, often leading to a form of programmed cell death [5–11].

NLRs belong to different phylogenetic clades carrying distinct N-terminal domains, with the coiled-coil (CC)-type being the most widespread in plants [12]. A canonical plant CC-NLR is Arabidopsis AtZAR1, which is sequestered in a resting monomeric state in complex with its partner pseudokinase RKS1. Perception of its cognate effector AvrAC primes individual AtZAR1 monomers, which undergo conformational changes and subsequently assemble into pentameric resistosomes [9,13]. Following resistosome assembly, CC-NLR oligomers insert themselves into the plasma membrane and presumably act as calcium channels, initiating immune signalling and cell death, as shown for AtZAR1 and the wheat CC-NLR TmSr35 [7,14]. In previous studies from our group, we showed that NbNRC2 and NbNRC4 form plasma membrane associated oligomers upon activation [15,16]. More recently, cryo-EM based structural studies showed that NbNRC2 and SlNRC4 form inflammasome-like resistosomes upon activation, although unlike AtZAR1 and TmSr35, these NRC assemblies are hexameric [17,18]. However, the extent to which the model of plant NLR activation drawn from CC-NLRs like AtZAR1 or TmSr35 which involves conversion from a monomeric resting state to an oligomer applies to other CC-NLRs, particularly to helper CC-NLRs, remains unknown.

NLR proteins have evolved from multifunctional singletons that can mediate both pathogen effector perception and subsequent immune signalling, as is the case for AtZAR1 [19,20]. However, several NLRs have subfunctionalised throughout evolution into sensors that are dedicated to pathogen perception and helpers that initiate immune signalling. NLR sensors and helpers can function together as receptor pairs or in higher-order configurations termed immune receptor networks [19]. In the NRC immune receptor network of solanaceous plants, multiple sensor CC-NLRs can signal redundantly via an array of downstream paralogous helper CC-NLRs termed NRCs (NLRs required for cell death) to confer disease resistance to a multitude of pathogens and pests [21]. Effector perception by NRC-dependent sensors leads to conformational changes which allow their central nucleotide-binding (NB) domain to signal to downstream helper NRCs and trigger their oligomerization into a helper resistosome [16,22,23]. Although assembly into pentameric or hexameric resistosomes is a shared endpoint between AtZAR1 and NRCs, respectively, the early stages of NRC activation are not understood. Whether the resting form of NRCs exhibits a similar conformation as the monomeric resting state AtZAR1 is not known.

In this work, we used cryo-EM to solve the structure of the helper NLR NbNRC2 purified from the model plant *Nicotiana benthamiana*. We show that unlike the canonical CC-NLR

AtZAR1, NbNRC2 forms a homodimer in its resting state. Mutagenesis of various residues in the NbNRC2 homodimerization interface led to impaired self-association and effector-independent activation, suggesting that homodimerization contributes to immune receptor autoinhibition. We leveraged our recent finding that the NB domains of sensor NLRs are sufficient to activate downstream NRC helpers [22] to study the activation of the NbNRC2 homodimer by the NB domain of the *Potato virus X* disease resistance protein Rx (Rx$^{NB}$). Using in vitro reconstitution assays, we show that Rx$^{NB}$ can trigger formation of a high molecular weight NbNRC2 oligomer. Remarkably, even though NbNRC2, NbNRC3, and NbNRC4 are genetically redundant paralogs for Rx-mediated signalling and disease resistance [21], NbNRC2 does not hetero-associate or co-localise with other NRC paralogs from *N. benthamiana*. These findings were corroborated with phylogenomic analysis of the homodimerization interface across the 15 major clades of NRC proteins found in the Solanaceae family. This interface has diverged within the majority of the individual NRC clades likely leading to molecular insulation of helper nodes in the immune receptor network. We propose that insulation of NRC helper pathways emerged throughout evolution to generate genetic redundancy, minimise undesired cross-activation, and evade pathogen suppression of immunity. Our work expands our understanding of the mechanisms of NLR activation, pointing to helper NLR conformational transitions from homodimers to oligomeric resistosomes.

## Results

### Cryo-EM structure of the NbNRC2 homodimer

Blue native polyacrylamide gel electrophoresis (BN-PAGE) suggests that the helper NLR NbNRC2 accumulates in planta at a higher molecular weight than a monomer [16,22–24]. This prompted us to determine whether this protein self-associates in planta in its resting state. Co-immunoprecipitation assays conducted in the solanaceous model plant *N. benthamiana* revealed that NbNRC2 self-associates, unlike the canonical functional singleton NbZAR1 (**Fig 1A**).

To further investigate the resting form of NbNRC2, we transiently expressed and affinity-purified NbNRC2 from leaves of *N. benthamiana*. Size-exclusion chromatography with this sample revealed elution volumes consistent with a helper NLR homodimer (**S1 Fig**). Subsequent negative-staining imaging of purified NbNRC2 and their class averages also suggested an NRC2 homodimer (**Fig 1B**). Cryo-EM imaging of these particles and single particle reconstruction revealed a homodimer structure at a resolution range of 3.9 to 4.5 Å with C2 symmetry imposed (**Figs 1C and S2, S3 and S4 and S1 Table**). The reconstructed structure contains the NB-ARC and LRR domains of NbNRC2, with the N-terminal CC domain appearing disordered in the density map. Flexible fitting of the AlphaFold2 model of NbNRC2 monomers into the cryo-EM map of the dimeric assembly revealed the relative orientation of the helper NLR protomers as well as the domains responsible for homodimer organisation (**Figs 1C and 1D and S3 and S4 and S1 Movie**).

The dimerization interface can be divided into 3 parts. The NB domain of each NbNRC2 protomer interacts with the LRR region of the other protomer, forming 2 similar interfaces related by 2-fold symmetry of the dimer (interface 1 and interface 3) (**Fig 1C**). The N-terminal LRR region of each NbNRC2 protomer stacks over the equivalent region of the other protomer, at the 2-fold symmetry, forming interface 2 (**Fig 1C**). These 3 interfaces hold the homodimer together with a buried surface area 937 Å$^2$ for each protomer, making the total buried surface 1874 Å$^2$. The residues at the dimerization interface were listed with a distance cut-off of 6 Å (**S2 Table**). Interfaces 1 and 3 are composed of 3 stretches of residues in the NB domain (Stretch 1a-1c, residues 217–220, 238–244, and 270–274, respectively) in 1 protomer and 3 stretches of

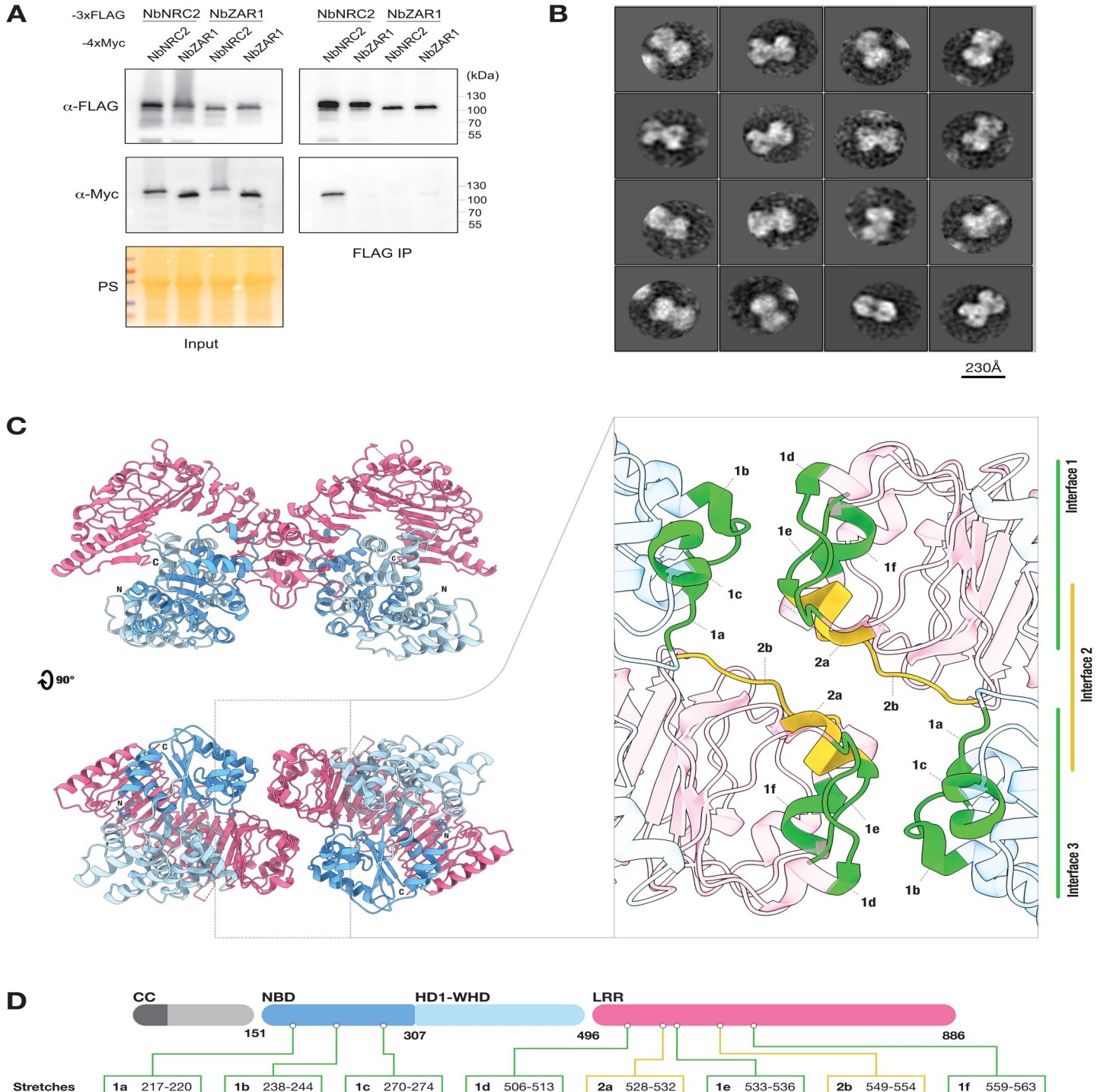

**Fig 1. The helper NLR NRC2 forms a homodimer in planta. (A)** Co-immunoprecipitation assays to test self-association of NRC2 and ZAR1 from *N. benthamiana*. C-terminally 3xFLAG-tagged NbNRC2 and NbZAR1 proteins were co-expressed with C-terminally 4xMyc tagged NbNRC2 or NbZAR1. Immunoprecipitations were performed with agarose beads conjugated to FLAG antibodies (FLAG IP). Total protein extracts were immunoblotted with the antisera labelled on the left. Approximate molecular weights (in kDa) of the proteins are shown on the right. Protein loading control was carried out using Ponceau stain (PS). The experiment was performed 3 times with similar results. **(B)** 2D classifications from negative staining transmission electron microscopy images of affinity-purified NbNRC2. NbNRC2 appears as a dimer, with a maximum dimension of approximately 150 Å at its widest point. Scale bar = 230 Å. **(C)** Cryo-EM structure of resting state of NRC2 dimers. Colour coding can be found in (**D**). Atomic model corresponding to NRC2 homodimer shown in 2 orthogonal views, with resolved NB domain (NBD), HD1-WHD, and LRR domains. Notably, the N-terminal CC domain of NbNRC2 was absent from the cryo-EM density. Inset shows details of interfaces between the 2 NbNRC2 protomers, highlighting amino acid stretches corresponding to three contact interfaces. **(D)** Colour coding of the domains is shown in the schematic representation of the domain architecture and

boundaries of NRC2, which includes the exact boundaries of the amino acid stretches at the dimerization interfaces. A schematic representation of the pipeline used for cryo-EM imaging, data processing and model building can be found in **S2 Fig**. Additional views of the structure and dimerization interface can be found in **S3** and **S4 Figs** and **S1 Movie**. Additional information on image processing and model building can be found in **S1 Table**. The full data underlying the structure in this figure can be found in the Protein Data Bank, PDB 8RFH.

## Homodimerization contributes to autoinhibition of resting state NbNRC2

In order to functionally validate our cryo-EM structure and to gain insights into the biological relevance of NbNRC2 homodimerization, we made 15 NbNRC2 variants with mutations in residues at the inter-protomer interaction interface. All 15 NbNRC2 variants generated accumulated in planta when transiently expressed in leaves of *nrc2/3/4* KO *N. benthamiana* (**S5 Fig**). Strikingly, 1 variant NbNRC2$^{E243R}$ became autoactive, triggering effector-independent cell death at 3 days post-infiltration (**Figs 2A** and **S6**). Next, we co-expressed all 15 NbNRC2 variants with Rx or with Rx/CP (**Figs 2A** and **S7**). This revealed 6 additional NbNRC2 variants that triggered effector-independent cell death when Rx was co-expressed. These were NbNRC2$^{C241A}$, NbNRC2$^{Y508A}$, NbNRC2$^{K512A}$, NbNRC2$^{P554E}$, NbNRC2$^{T560F}$, and NbNRC2$^{K563E}$, which we termed sensitised/trigger-happy [25]; 6 out of 7 mutations that led to autoactive or trigger-happy phenotypes were in residues found in interfaces 1 and 3, mediated by NB to LRR inter-protomer interactions. NbNRC2$^{P554E}$ was the only mutant in the LRR-LRR interface 2 region with a trigger-happy phenotype. All variants triggered cell death when co-expressed with Rx and CP except for NbNRC2$^{D239K}$, which did not trigger cell death upon Rx activation. Residue D239 has also been shown to mediate inter-protomer contacts in the active NbNRC2 hexameric resistosome, so the absence of cell death in the NbNRC2$^{D239K}$ may be due to compromised resistosome assembly [17]. We next assessed all 7 autoactive or trigger-happy NbNRC2 variants in co-immunoprecipitation assays to determine whether they were impaired in self-association. Tissue was harvested at 2 days post infiltration, before the onset of visible cell death triggered by NbNRC2$^{E243R}$. Self-association was abolished in all variants tested, except for variant NbNRC2$^{K563E}$ which displayed weak self-association compared to WT NbNRC2 (**Fig 2B**).

## The NbNRC2 homodimer converts into a resistosome upon activation

To further study NRC2 activation, we set up in vitro reconstitution experiments with purified NbNRC2 and carried out BN-PAGE assays. First, to determine the optimal experimental conditions, we incubated purified NbNRC2 with increasing levels of ATP, which is necessary for NLR activation [4,26]. Incubation of NbNRC2 with 15 mM ATP led to the disappearance of the lower molecular weight NRC2 band and appearance of higher molecular weight complexes migrating at a similar size to the NbNRC2 oligomer observed in planta [16,22–24] (**S8 Fig**). This indicates that purified NbNRC2 may get spontaneously activated at 15 mM ATP in vitro. We selected 5 mM ATP for subsequent assays.

Next, we hypothesized that the disease resistance protein Rx disrupts the NbNRC2 homodimer to trigger activation of its helper NLR. To test this, we used Rx$^{NB}$, the 154 amino acid NB

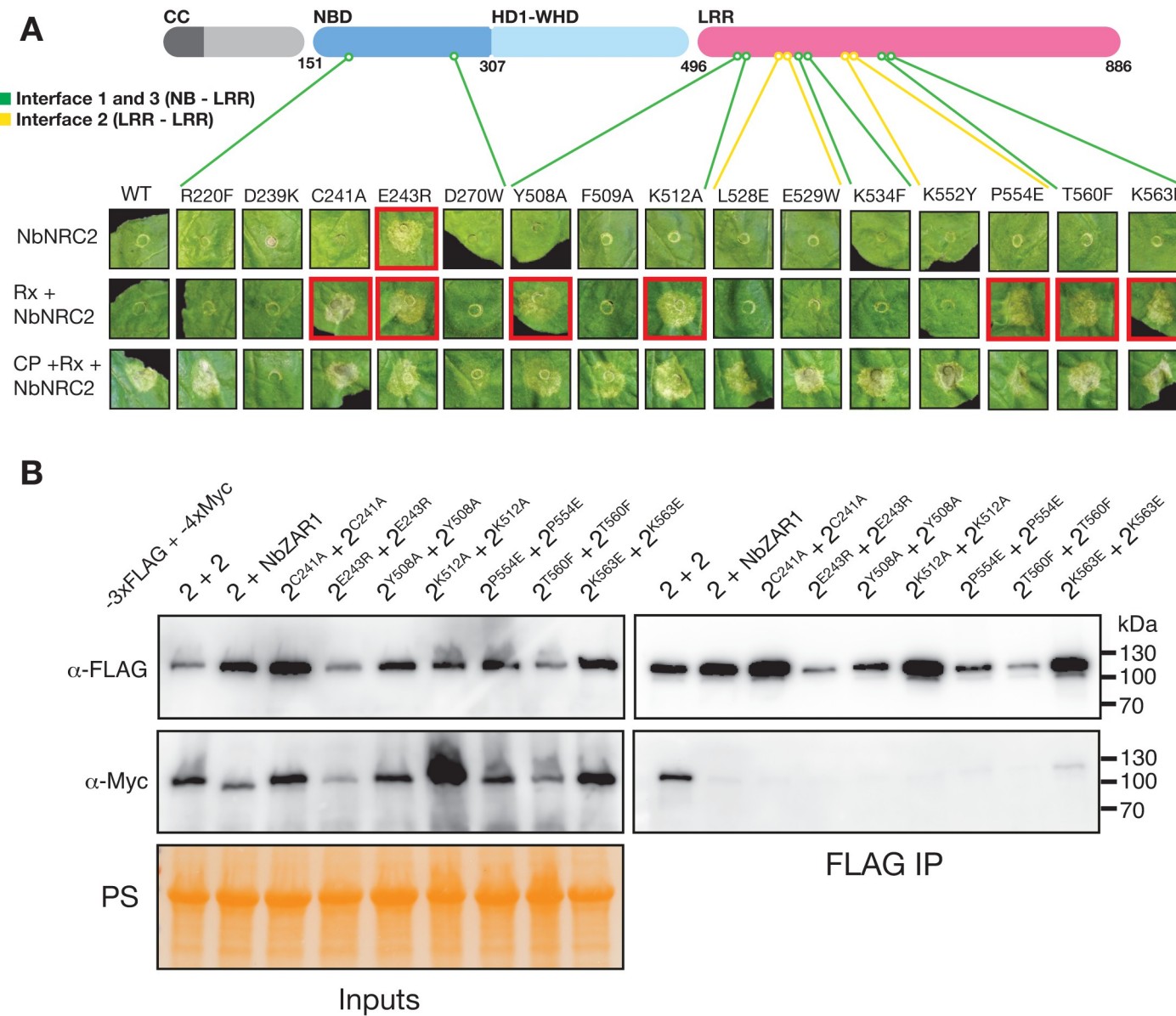

**Fig 2. Homodimerization contributes to autoinhibition of resting state NbNRC2.** Mutagenesis of amino acids found at the NbNRC2 homodimerization interface can lead to autoactivity and impaired self-association. (**A**) Schematic representation of NbNRC2 domain organisation, indicating the approximate position of the mutagenized residues. Green lines indicate residues corresponding to interfaces 1 and 3 (NB domain to LRR domain inter-protomer interface) and yellow lines indicate residues corresponding to interface 2 (LRR to LRR inter-protomer interface). Cell death assays were carried out by transiently expressing each NbNRC2 variant in leaves of *nrc2/3/4* KO *N. benthamiana* either on its own (top row), together with the sensor NLR Rx (middle row) or with the sensor NLR Rx and PVX CP (bottom row). Red outline on boxes indicate effector-independent activation (autoactivation or trigger-happy phenotypes). Quantification of these cell death assays can be found in **S6 and S7 Figs**. (**B**) Co-immunoprecipitation assays to test self-association of NbNRC2 and 7 NbNRC2 homodimerization interface mutants with impaired autoinhibition. For each treatment, C-terminally 3xFLAG-tagged NbNRC2 variants indicated (labelled 2) were co-expressed with a C-terminally 4xMyc tagged version of the same variant (also labelled 2). WT NbNRC2 self-association was included as a positive control, and association between NbNRC2 and NbZAR1 was included as a negative control. Immunoprecipitations were performed with agarose beads conjugated to FLAG antibodies (FLAG IP). Total protein extracts were immunoblotted with the antisera labelled on the left. Approximate molecular weights (in kDa) of the proteins are shown on the right. Protein loading control was carried out using Ponceau stain (PS). The experiment was performed three times with similar results.

domain of Rx, which is necessary and sufficient to activate NbNRC2 in the absence of a pathogen effector [22,27]. In the presence of 5 mM ATP, these in vitro reconstitution experiments revealed that increasing amounts of Rx$^{NB}$ triggered the appearance of higher order complexes

similar in size to the NbNRC2 resistosome (**S8 Fig**). We conclude that conversion of the helper NLR homodimer into a resistosome by a sensor NLR is a key step of NbNRC2 activation in a mechanistic model that is different from the activation of the ZAR1/RKS1 complex [9,13].

## NRC paralogs do not associate with each other

Paralogous NRC helpers can form genetically redundant network nodes in the ~100-million-year-old immune receptor network, therefore promoting robustness and evolvability of the plant immune system [3,21,28]. However, the biochemical basis of NRC redundancy is unknown. We hypothesized that NRC paralogs acquired mutations in their dimerization interfaces that insulate them from one another, as predicted from prior theory on the evolution of orthogonal signalling pathways following gene duplication events [29,30]. We investigated the degree to which NRC paralogs form heterodimers with each other using in planta co-immunoprecipitation assays of resting forms of NbNRC2, NbNRC3, and NbNRC4, and used NbZAR1 as a negative control. Our assays confirmed NbNRC2 self-association but yielded weak to non-detectable signal for hetero-association between NbNRC2 and NbNRC3, NbNRC4 or NbZAR1 (**Fig 3A**). We conclude that NbNRC2 preferentially self-associates in *N. benthamiana* and does not form stable heterocomplexes with its paralogs NbNRC3 and NbNRC4.

We independently corroborated these findings using live-cell confocal microscopy (**Fig 3B**). NbNRC2 accumulates in filament-like structures in its resting state that do not over-lap with the previously reported peripheral subcellular localization of NbNRC4 [15] further supporting the view that these critical network nodes signal via insulated pathways. Overall, these findings are consistent with the distinct sequences of the dimer interfaces of NRC helper NLRs inferred from the NbNRC2 homodimer structure (**Fig 3C** and **S1 Data**).

## Homodimerization interfaces have diverged throughout NRC evolution

The observation that NbNRC2 does not form heterocomplexes with other NRC paralogs despite their common evolutionary origin prompted us to perform phylogenomic analyses across the highly expanded NRC clade of the Solanaceae family. First, we developed a computational pipeline to extract 1,092 NRC sequences from 6,630,292 predicted proteins from 123 solanaceous genome assemblies representing 38 species of the genera *Solanum*, *Physalis*, *Capsicum*, and *Nicotiana* (**S2 Data**). We grouped these NRC sequences into 15 phylogenetic clades ranging in size from 31 to 129 sequences (**S9 Fig**), and 10 of the 15 clades include orthologs from at least 2 genera, therefore representing relatively deep NRC lineages (**S10 Fig**).

The consensus sequences of the 8 amino acid stretches found in the dimerization interface were, overall, degenerated across all NRC sequences compared to known conserved motifs of CC-NLRs, e.g., P-loop and MHD motifs, suggesting that they have diversified throughout NRC evolution (**Figs 4A** and **S10** and **S3 Data**). However, the dimerization interfaces tended to be conserved within each clade (**Figs 4B** and **S10** and **S11** and **S4 Data**). We identified unique polymorphisms within the major NRC2, NRC3, and NRC4 clades by calculating the relative amino acid ratio for one clade versus the others using a 50-fold cutoff with a minimum frequency of 80% within the query clade (**Fig 3B**). Using these criteria, the NRC2 and NRC3 clades, which contain representative proteins used in co-immunoprecipitation assays in **Fig 3A**, had 5 and 3 unique polymorphisms among the 44 amino acids that map to the homodimer interface (**Fig 4B** and **S5 Data**).

Next, we corroborated these findings using Shannon's entropy analyses. The entropy of a given amino acid position ranges from zero ($Log_2 1$) for fully conserved residues to a

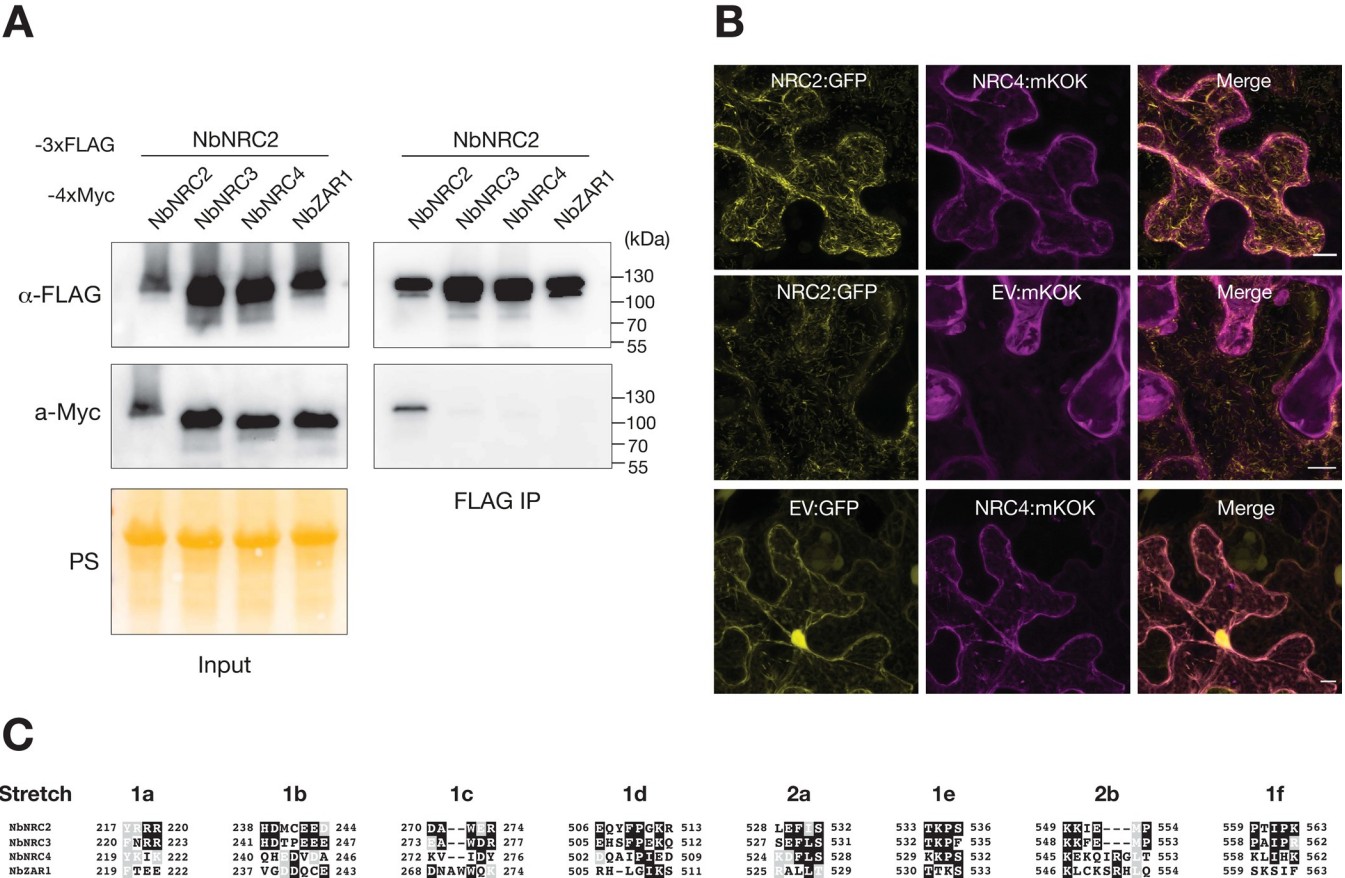

**Fig 3. NRC2 does not hetero-associate or co-localise with other NRC paralogs in its resting state.** (**A**) Co-IP assays to test the association of NRC2 with its paralogs NRC3 and NRC4 from *N. benthamiana*. C-terminally FLAG-tagged NbNRC2 was co-expressed with C-terminally 4xMyc-tagged NbNRC2, NbNRC3, NbNRC4, or NbZAR1 as a negative control. IPs were performed with agarose beads conjugated to FLAG antibodies (FLAG IP). Total protein extracts were immunoblotted with the antisera labelled on the left. Approximate molecular weights (kDa) of the proteins are shown on the right. Protein loading control was carried out using Ponceau stain (PS). The experiment was performed 3 times with similar results. (**B**) Z-stacks of confocal micrographs show the localization of NbNRC2:GFP together with NbNRC4:mKOk or EV:mKOk. C-terminally GFP-tagged NbNRC2 was co-infiltrated with either NbNRC4:mKOk or EV:mKOK in wild-type *N. benthamiana* plants. EV:GFP was co-infiltrated with NbNRC4:mKOk as an additional control. Scale bars represent 10 μm. Images were taken 48 h post-agroinfiltration. (**C**) Amino acid sequence alignment of the 8 stretches identified at the NbNRC2 dimerization interface, comparing NbNRC2, NbNRC3, NbNRC4, and NbZAR1.

maximum of 4.32 (Log$_2$20) when all 20 amino acids are equally found (30). We used a threshold of 1.5 to highlight highly variable positions consistent with previous NLR studies (31). All 8 dimerization stretches scored above 1.5 when all 1,092 NRC helpers were compared but were highly conserved within 11 of the 15 NRC clades (**Figs 4A** and **S10** and **S11**). In 6 clades (NRC0, NRCX, NRC2, NRC3, NRC4a, and NRC4t), all residues across the 8 stretches scored under the 1.5 threshold and showed little to no sequence diversity (**S10** and **S11 Figs** and **S4 Data**). In 5 clades (NRC1, NRC6, NRC7, NRC4b, and NRC4$_{other}$), the 8 stretches were under the 1.5 threshold except for 1 to 3 variable residues within each clade (**S10 Fig**). In the remainder of the clades (NRC8, NRC9, NRC5, and NRC4c), all 8 stretches had up to 12 variable residues within each clade with entropy values higher than 1.5 indicating higher levels of sequence diversity compared to other clades (**S4 Data**). In contrast, conserved motifs, such as P-loop and MHD, were consistently under the defined threshold with 13 of the 15 clades scoring under 1.5 and displaying little to no sequence variation (**Figs 4** and **S10** and **S11** and **S4 Data**).

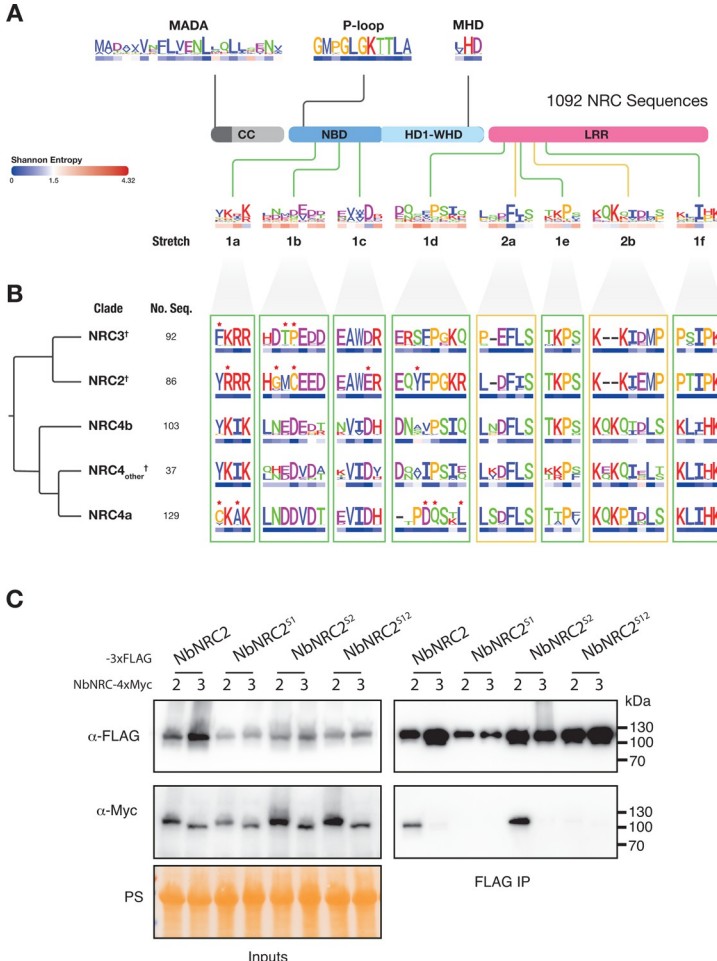

**Fig 4. The NRC homodimerization interface has diversified throughout evolution of NRC clade.** (**A**) Schematic representation of NRC domain architecture, featuring CC, NBD, HD1-WHD, and LRR domains. Approximate position of different previously characterized NLR motifs (MADA, p-loop, MHD) and the 8 amino acid stretches present at the dimerization interface are indicated. A database of 993 unique NRC sequences and 99 reference NRC sequences was compiled from a total of 123 solanaceous genomes (**S2 Data**) and used to generate consensus sequence patterns for each motif/region (**S3 Data**). Shannon entropy scores for each amino acid position were calculated using this database and indicated, in bits, below each position in the consensus sequence pattern (**S4 Data**). A Shannon entropy score of 1.5 was established as a threshold to distinguish between highly variable (>1.5 bits) and non-highly variable (<1.5 bits) amino acids, as established previously [31]. (**B**) All 1,092 NRC sequences were grouped by phylogeny (**S8 Fig**), and consensus sequence patterns within each clade were generated with sequence logos for each of the 8 amino acid stretches that define the dimerization interface (**S9 Fig**). NRC clades are ordered by phylogenetic relationship, indicated on the left along with the number of NRC sequences within each clade. Shannon entropy score is indicated below each amino acid in the consensus sequence pattern. Red stars above indicate amino acid polymorphisms that are unique to each clade, as determined by analysing their relative amino acid ratio among NRC clades (**S5 Data**). †Indicates clades containing *N. benthamiana* NRCs used in the co-immunoprecipitation experiments in **Fig 3A**. More detailed Shannon entropy plots calculated across all 1,092 NRC sequences and within each individual NRC clade are shown in **S10 Fig**. (**C**) Co-immunoprecipitation assays to test self-association of NbNRC2 or NbNRC3 with various NbNRC2 variants carrying the homodimerization interface of NbNRC3. We generated swaps at the NB to LRR inter-protomer interface (interfaces 1 and 3, comprised of stretches 1a-f and referred to as NbNRC2^S1), at the LRR to LRR inter-protomer interface (interface 2, comprised of stretches 2a-b, NbNRC2^S2) and at both (NbNRC2^S12). C-terminally 3xFLAG-tagged NbNRC2, NbNRC2^S1, NbNRC2^S2, and NbNRC2^S12 were coexpressed with C-terminally 4xMyc tagged NbNRC2 or NbNRC3. Immunoprecipitations were performed with agarose beads conjugated to FLAG antibodies (FLAG IP). Total protein extracts were immunoblotted with the antisera labelled on the left. Approximate molecular weights (in kDa) of the proteins are shown on the right. Protein loading control was carried out using Ponceau stain (PS). The experiment was performed two times with similar results.

To determine the extent to which residues at the homodimerization interface contribute to insulation between NRC paralogs, we swapped residues at the homodimerization interface from NbNRC3 into NbNRC2 (Fig 3C) and tested these chimeric proteins in co-immunoprecipitation assays (Fig 4C). We generated swaps at the NB to LRR inter-protomer interface (interfaces 1 and 3, comprised of stretches 1a-f and referred to as NbNRC2$^{S1}$), at the LRR to LRR inter-protomer interface (interface 2, comprised of stretches 2a-b, NbNRC2$^{S2}$) and at both (NbNRC2$^{S12}$). NbNRC2$^{S1}$ and NbNRC2$^{S12}$ did not associate with NbNRC2. Interestingly, NbNRC2$^{S2}$ associated with NbNRC2. None of the swaps tested gained the capacity to associate with NbNRC3. These results suggest that introducing NbNRC3 residues at the NB to LRR inter-protomer interface (S1) prevents association with NbNRC2, indicating that this interface likely mediates NRC inter-paralog insulation.

## Discussion

### Contrasting activation mechanisms across plant NLRs

To date, our understanding of the structural basis of the resting state of plant NLRs has been limited to that of AtZAR1 [13]. In this study, we show that a CC-NLR helper in the NRC immune network exhibits a distinct mechanism of activation relative to the ZAR1/ZRK system (S12 Fig). Our data is consistent with a model in which exposure of the NB domain of the disease resistance protein Rx converts the homodimer of its downstream helper NbNRC2 into a high molecular weight oligomer. The cryo-EM structure of the NbNRC2 homodimer revealed novel intermolecular interactions that hold this helper NLR in an autoinhibited resting state, which can lead to effector-independent activation when mutagenized. The surfaces involved in NbNRC2 homodimerization are surface exposed in the activated NbNRC2 hexamer structure [17]. This implies that conversion from homodimer to a multimeric resistosome is likely to involve dimer dissociation into monomers or large-scale conformational rearrangements in the dimer, likely with additional intermediate assemblies. Further work will shed light on the precise dynamics underpinning transition from homodimer to higher-order oligomers. In particular, how sensor NLRs trigger homodimer activation and whether primed monomers precede NRC oligomerization remain open questions.

The contrasting models in early steps of activation of NbNRC2 and AtZAR1 indicate that plant NLRs have evolved a diversity of activation and autoinhibition mechanisms possibly associated with their evolutionary transitions from functional singletons to paired and networked immune receptors. Moreover, in another work, we showed that Rx/CP-activated NbNRC2 forms hexameric oligomers, unlike the pentameric assemblies formed by activated AtZAR1, pointing to differing models both at resting state and following activation [17]. Another recently published study on the constitutively active SlNRC4 variant SlNRC4$^{D478V}$ revealed that it also forms hexameric assemblies with calcium channel activity, suggesting that hexamerization applies to other NRCs as well [18]. We propose that AtZAR1 is more likely to reflect the ancestral state of CC-NLRs given that it is present across ancient plant lineages, including dicotyledons, monocotyledons, and magnoliids and has been proposed to guard ZRK proteins since the Jurassic era [32,33]. In contrast, NRC sensor/helper pairing is thought to be a more recent innovation, having emerged in the Caryophyllales and Asterid lineages of dicotyledoneous flowering plants [21,34,35]. It will be interesting to determine the extent to which homodimerization has evolved as a steady state across NLRs. Already, other NLRs including RPM1, a CC-NLR outside the NRC superclade, have been known to self-associate in its resting state, suggesting that resting state self-association is not exclusive to paired or networked NLRs [36,37].

More recently, Ma and colleagues showed that SlNRC2, the *Solanum lycopersicum* ortholog of NbNRC2, also forms homodimers [38]. Interestingly, they report that resting state SlNRC2 homodimers can assemble into tetramers and filaments similar to the mobile filamentous structures identified by Duggan and colleagues using live cell imaging [15,38]. While the SlNRC2 homodimer also appears to be contributing towards helper autoinhibition, mutations in the tetramerization or filament formation interfaces did not lead to effector independent activation, making the biological function of these assemblies unclear [38]. Interestingly, NRC4 does not form filaments at the resting state and instead accumulates at the plasma membrane, shifting to the interface between *Phytophthora infestans* and the host plant at the site where effectors are delivered before re-localising and forming discrete puncta at the plasma membrane following activation [15]. In the future, we need a better understanding of the diversity of resting state oligomeric complexes and subcellular assemblies of NRC helpers and the biological activities associated with this diversity.

How does homodimerization contribute to sensor-helper communication? Interestingly, while the NbNRC2$^{E243R}$ variant triggered effector independent cell death even in the absence of sensor NLR overexpression, several NbNRC2 homodimerization interface mutants tested became autoactive only when the sensor NLR Rx was overexpressed. This suggests that homodimerization also prevents helper mis-activation by resting state sensor NLRs. The observation that NbNRC2 mutants impaired in homodimer formation can spontaneously activate when high doses of resting state sensor NLR are present suggests that sensors and helpers may be capable of engaging in transient, low affinity, interactions even prior to effector perception. We hypothesize that in the case of NbNRC2 these interactions would be of insufficient affinity or stability to lead to helper activation and that when NbNRC2 homodimer stability is compromised, high doses of resting state sensor become sufficient to trigger activation.

Our findings that mutations that compromise homodimerization display effector-independent autoactivity has implications for bioengineering boosted helper NLRs. While several NbNRC2 homodimerization mutants triggered strong cell death upon Rx overexpression, they were not autoactive when expressed in a cell with normal doses of endogenous sensors (**S6 Fig**). It is possible that these homodimerization interface mutants will respond to lower doses of effector-activated sensors, therefore leading to enhanced immune activation dynamics and improved disease resistance. Previous work by Segretin and colleagues identified mutations in the CC-NLR R3a that led to enhanced cell death upon effector perception, also termed sensitised or "trigger-happy" variants [25]. While the precise mechanistic basis for this trigger-happy phenotype in R3a is not known, it is possible that it is due to the disruption of autoinhibitory interactions as is the case for our NbNRC2 homodimerization interface mutants. Whether R3a or other NLRs also form resting state autoinhibitory dimers or oligomers remains to be tested. Nonetheless, we hypothesize that making mutations to alter resting state NLR immune receptor autoinhibition may provide an exciting strategy to bioengineer enhanced disease resistance.

NLRs are generally thought to be hypervariable due to coevolution with rapidly evolving pathogens [39,40]. While the NRC clade of helper NLRs exhibits greater conservation within the Solanaceae compared to NRC-dependent sensors and most other NLRs [21,34,35], these proteins exhibit variable regions across the NRC helper clade (**Figs 4** and **S10 and S11**). Our analyses revealed that distinct NRC clades have diverged from each other in the homodimerization interface and that NRC2 does not associate with its NRC3 and NRC4 paralogs. Swapping the NbNRC3 homodimerization interface into NbNRC2 led to NbNRC2 variants that failed to associate with NbNRC2. We propose that following duplication, NRC paralogs have accumulated mutations in the dimerization interface that prevent them from forming heterodimers, thereby forming insulated pathways in the immune network. The established paradigm is that sequence diversification of plant NLRs is primarily driven by co-evolutionary arms-races with

rapidly evolving pathogens. Here, we show that NRC helpers have rapidly diversified, potentially to prevent hetero-association and form insulated redundant signalling nodes, expanding our understanding of the adaptive drivers of NLR diversification. However, co-evolution with pathogens may have also driven the emergence of insulated NRC pathways given that redundant nodes in this receptor networks may enable the plant to evade suppression by pathogen effectors, contributing to the robustness of the immune system (**S12 Fig**) [20,24,41,42].

## Materials and methods

### Plant growth conditions

Wild-type and *nrc2/3/4* CRISPR mutant [43] *N. benthamiana* lines were grown in a controlled environment growth chamber with a temperature range of 22 to 25°C, humidity of 45% to 65% and a 16/8-h light/dark cycle.

### Plasmid construction

We used the Golden Gate Modular Cloning (MoClo) kit [44] and the MoClo plants part kit [45] for cloning. All vectors used were generated with these kits unless otherwise stated. Cloning design and sequence analysis were done using Geneious Prime (v2021.2.2; https://www.geneious.com). All NRC and ZAR1 constructs used for co-immunoprecipitation assays were cloned into the pICH86988 level one acceptor with a C-terminal 3xFLAG tag (pICSL5007) or a C-terminal 4xMyc tag (pICSL5010). Constructs for protein purification were cloned into pJK268c-mRFP acceptor with 2 × 35 s promoter (pICSL51288) and terminator (pICSL41414). NbNRC2 was cloned with a C-terminal 6xHis-3xFLAG tag. $Rx^{NB}$-eGFP and eGFP constructs were cloned with an N-terminal 6xHis-3xFLAG tag. Constructs used for cell biology were previously reported [15,16].

### Transient protein expression by agroinfiltration in *Nicotiana benthamiana*

Proteins of interest were transiently expressed in *N. benthamiana* according to previously described methods [16]. Briefly, leaves from 4- to 5-week-old plants were infiltrated with suspensions of *Agrobacterium tumefaciens* GV3101 pM90 strains transformed with expression vectors coding for different proteins indicated. Final $OD_{600}$ of all *A. tumefaciens* suspensions were adjusted in infiltration buffer (10 mM MES, 10 mM $MgCl_2$, and 150 μm acetosyringone (pH 5.6)). Final $OD_{600}$ used for each construct was 0.3.

### Cell death assays by agroinfiltration in *Nicotiana benthamiana*

Strains were transformed with expression vectors coding for different proteins indicated. Final $OD_{600}$ of all *A. tumefaciens* suspensions were adjusted in infiltration buffer (10 mM MES, 10 mM $MgCl_2$, and 150 μm acetosyringone (pH 5.6)). Final $OD_{600}$ used was 0.3 for each NLR construct and 0.1 for CP-eGFP or eGFP. Whenever multiple constructs were co-infiltrated into an individual spot, the total concentration of bacteria was kept constant across treatments by adding untransformed *A. tumefaciens* when necessary. This was to avoid an effect from differences in the total $OD_{600}$ of bacteria in each treatment.

### Co-immunoprecipitation (co-IP) assays

Co-immunoprecipitation assays were performed as described previously [41]. Four- to five-week-old *N. benthamiana* plants were agroinfiltrated as described above with constructs of interest and leaf tissue was collected 3 days post agroinfiltration. Final $OD_{600}$ used for each construct was 0.3. Leaf tissue was ground using a Geno/Grinder tissue homogenizer. GTEN

extraction buffer was used (10% glycerol, 50 mM Tris-HCl (pH 7.5), 5 mM $MgCl_2$, and 50 mM NaCl) supplemented with 10 mM DTT, 1× protease inhibitor cocktail (SIGMA) and 0.2% IGEPAL (SIGMA). Samples were incubated in extraction buffer on ice for 10 min with short vortex mixing every 2 min. Following incubation, samples were centrifuged at 5,000 x*g* for 15 min and the supernatant was collected. This was spun down an additional time at 5,000 *xg* for 15 min and then supernatant was filtered using Minisart 0.45 μm filter (Sartorius Stedim Biotech, Goettingen, Germany).

Part of the extract was set aside prior to immunoprecipitation. These were used as inputs. 1.4 ml of the remaining filtered total protein extract was mixed with 30 μl of anti-FLAG agarose beads (SIGMA) and incubated end over end for 90 min at 4˚C. Beads were washed 5 times with immunoprecipitation wash buffer (GTEN extraction buffer with 0.2% v/v IGEPAL (SIGMA)). Associated plant proteins were competitively eluted by excess of 3xFLAG peptides. Elution was spun down at 1,000 *xg* for 1 min and the supernatant was transferred to a new tube. Inputs and eluted immunoprecipitates were diluted in SDS loading dye and denatured by heating for 10 min at 72˚C. All samples were used for SDS-PAGE. Briefly, they were run on 4% to 20% Bio-Rad Mini-PROTEAN TGX gels alongside a PageRuler Plus prestained protein ladder (Thermo Scientific). The proteins were then transferred to polyvinylidene difluoride membranes using Trans-Blot Turbo Transfer Buffer using a Trans-Blot Turbo Transfer System (Bio-Rad) as per the manufacturer's instructions. Membranes were immunoblotted as described below.

### Immunoblotting and detection of BN-PAGE and SDS-PAGE assays

Blotted membranes were blocked with 5% milk in Tris-buffered saline plus 0.01% Tween 20 (TBS-T) for an hour at room temperature and subsequently incubated with desired antibodies at 4˚C overnight. Antibodies used were anti-GFP (B-2) HRP (Santa Cruz Biotechnology), anti-FLAG (M2) HRP (Sigma), or anti-Myc (9E10) HRP (Roche), as indicated in the figures, all used in a 1:5,000 dilution in 5% milk in TBS-T. To visualise proteins, we used Pierce ECL Western (32106, Thermo Fisher Scientific), supplementing with up to 50% SuperSignal West Femto Maximum Sensitivity Substrate (34095, Thermo Fisher Scientific) when necessary. Membrane imaging was carried out with an ImageQuant LAS 4000 or an ImageQuant 800 luminescent imager (GE Healthcare Life Sciences, Piscataway, New Jersey, United States of America). Rubisco loading control was stained using Ponceau S (Sigma) or Ponceau 4R (A.G. Barr).

### Protein purification from *N. benthamiana*

Around 30 leaves of *nrc2/3/4* KO *N. benthamiana* were agroinfiltrated as described above to transiently express NbNRC2-6xHis-3xFLAG, 6xHis-3xFLAG-Rx[NB]-eGFP, or 6xHis-3xFLAG-eGFP. All constructs were cloned in pJK268c acceptor. Tissue was harvested after 3 days and snap-frozen in liquid nitrogen. All samples were stored at −80˚C until the protein was finally extracted. The entire purification process was completed in the same day for each preparation. On the day of purification, the frozen tissue was ground into fine powder in a mortar and pestle that was pre-cooled with liquid nitrogen, and 20 g of ground powder were resuspended with extraction buffer (100 mM Tris-HCl (pH 7.5), 150 mM NaCl, 1 mM $MgCl_2$ 1 mM EDTA, 10% glycerol, 10 mM DTT, 1× cOmplete EDTA-free protease inhibitor tablets (Sigma) and 0.4% v/v IGEPAL). Approximately 20 g of ground tissue was resuspended in ice-cold extraction buffer at a 1:4 w/v ratio (20 g of powder in 80 ml of extraction buffer). After vortexing and resuspending the powder in this buffer, the crude extract was spun down for 10 min at max. speed. The supernatant was transferred to a new tube and centrifuged again for 10 min at max.

speed. This second supernatant was filtered using Miracloth (Merck), and 200 ml of anti-FLAG beads were added to the supernatant and incubated at 4°C for 90 min. The tubes were in constant rotation to prevent the beads from sedimenting. The protein-bound beads were collected on an open column and washed with 10 ml of wash buffer (100 mM Tris-HCl (pH 7.5), 150 mM NaCl, 1 mM $MgCl_2$ 1 mM EDTA, 10% glycerol, and 0.4% v/v IGEPAL). The washed beads were collected in an Eppendorf tube and eluted with the final isolation buffer in 200 ml volume (100 mM Tris HCl (pH 7.5), 150 mM NaCl, 1 mM MgCl2, 1 mM EDTA, protease inhibitor cocktail, 5% glycerol, and 0.2% IGEPAL) supplemented with 0.3 mg/ml 3xFLAG peptide. The eluted protein was used for SDS-PAGE to assess sample quality and purity. About 1.2 to 1.5 mg/ml concentration was obtained for NRC2, $Rx^{NB}$-eGFP, and eGFP from each purification, as determined by absorption at 280 nm. FLAG-eluted pure protein samples were used for all in vitro experiments and structural studies.

## Gel filtration assays with purified NbNRC2

C-terminally 6xHis-3xFLAG NbNRC2 was transiently expressed in leaves of nrc2/3/4 KO *N. benthamiana* as described above. C-terminally 6xHis-3xFLAG-tagged NbNRC2 used in gel filtration assays was cloned in pICH86988 acceptor. Extracts were purified as described above and injected onto Superdex 200 Increase 10/300 GL column (GE Healthcare) connected to an AKTA FPLC system (GE Healthcare). The column was pre-equilibrated in buffer containing 25 mM Tris-HCL (pH 8), 500 mM NaCl, 5% glycerol, and 1 mM EDTA. Samples were centrifuged prior to loading. The sample was injected at a flow rate of 0.2 ml/min at 4°C and 0.5 ml fractions were collected for analysis by SDS-PAGE.

## Negative staining

A total of 3 ml of purified NRC2 sample diluted to 0.3 mg/ml were applied to 400 mesh Copper grids (Agar Scientific) with continuous carbon that was glow discharged using a PELCO Easiglow (Ted Pella) for 30 s at 8 mA. After 30 s, the sample was blotted using Sartorius 292 filter paper and immediately washed with 2 consecutive drops of 100 ml distilled water each. After washing, 10 ml of 2% (w/v) uranyl acetate stain in $H_2O$ was applied. After 20 s, the excess was blotted off. Immediately after this, another 10 ml of 2% (w/v) uranyl acetate stain in $H_2O$ was applied for 30 s and then blotted off. The grid was air-dried and examined in a FEI Talos F200C transmission electron microscope (Thermo Fisher Scientific), operated at 200 keV, equipped with a 4k OneView CMOS detector (Gatan), and 15 representative micrographs were recorded with a magnified pixel size 1.7Å (**Fig 1B**) each with a high particle density. Particles were auto-picked from these 15 micrographs using RELION 3.1 without any template. 2D-classification of the auto-picked particles (2,447 particles) from these images revealed particles resembling putative NRC2 as dimers (**Fig 1B**).

## Cryo-EM sample preparation and data collection

Schematic representation of the Cryo-EM structure determination workflow can be found in **S2 Fig**. C-flat grids (Protochips: 1.2/1.3 300 mesh) were used for Cryo-EM grid preparation, and 3 ml of 1.2 mg/ml of full-length NRC2 sample were applied 2 times over negatively glow discharged C-flat grids and coated with graphene oxide. The sample was applied inside the chamber of a Vitrobot Mark IV (Thermo Fisher Scientific) at 4°C and 90% humidity and vitrified in liquid ethane. Without graphene oxide coating, no particles were observed in vitreous ice, so graphene-coated grids were used. The Cryo-EM images were collected on a FEI Titan Krios (Thermo Fisher Scientific) operated at 300 keV equipped with a K3 direct electron detector (Gatan) after inserting an energy filter with a slit width of 20 eV. Micrographs were

collected with a total dose of 50 e$^-$/Å$^2$, nominal magnification of 105 kx, giving a magnified pixel size of 0.828 Å. Images were collected as movies of 50 fractions with defocus values ranging from −1.5 μm to −2.7 μm with 2 exposures per hole. A total of 6,135 movies were collected for the image processing and 3D reconstruction using RELION 4.0 [46].

The micrograph movies were imported into RELION and subjected to drift correction using MotionCor2 [47] and CTFFIND4.0 was used for fitting of contrast transfer function and defocus estimation. The Laplacian-of-Gaussian auto-picker in RELION 4.0 was used for automatic reference-free particle picking [46]. The particles were extracted in 256-pixel box and subjected to several rounds of 2D classification with a circular mask of 150 Å. The 2D classes revealed a clear dimer with secondary structural features in many views. Clean 2D-classes were selected (664,305 particles) and were subjected to 3D classification using an ab initio 3D model generated by RELION. The best 3D class revealing the protein fold consisted of 229,347 particles and was subjected to 3D refinement using Refine3D using that 3D class as a reference with a circular mask of 180 Å. Following examination of the C1 refined map, a C2 symmetry was applied. After particle polishing, CTF refinement and postprocessing, the final average resolution was 3.9 Å as estimated by the Fourier shell correlation (FSC = 0.143). For post-processing, a solvent mask covering all protein volume was created and was extended by 8 pixels, to which a smooth soft-edge of 11 pixel were added. This mask was visually examined on how it covers the 3D-volume before post-processing to ensure that it is smooth and covers all the volume. The local resolution plot was calculated using PHENIX 3.0 [48]. The final map revealed distinct domain boundaries of the NRC2 protein fold with clear secondary structural features, helical twists, and individual beta strands and allowed us to place each NRC2 protomer and initiate model building at this resolution. A significant number of particle views are down the short dimension, owing to graphene oxide backing. The final map revealed the NB-ARC and LRR domains of the NRC2 molecule, with the N-terminal CC domain not resolved.

### Model building and refinement

The AlphaFold2 model of NRC2 monomer was divided into NB, ARC, and LRR domains. Each of these domains was fitted separately into the Cryo-EM density using UCSF ChimeraX as rigid body units [49]. Significantly, AlphaFold2 multimer was unable to predict the quaternary structure we observed experimentally. The model was then manually adjusted in COOT, combined and refined using the PHENIX real space refinement module and the fit of the atomic model to the features in the Cryo-EM map was evaluated visually (**S3 Fig**) [48,50]. This model was validated using PHENIX and MolProbity [48,51]. Figures were made using ChimeraX. The interface residues and the buried surface area were evaluated using PyMol and the PISA server [52,53]. A distance cut-off 6 Å is used to define residues at the interface to accommodate all short- and long-range interacting residues. These residues compose the 8 dimerization stretches indicated in **Fig 1C** and listed in **S2 Table**. These are the stretches carried for subsequent phylogenetic analyses. Further details on Cryo-EM data processing statistics can be found in **S1 Table**.

### In vitro NRC2 activation assays

To understand the impact of excess ATP on NRC2, 6 ml of 10 mM NbNRC2 were incubated with increasing concentration of ATP (0, 5, 10, and 15 mM, respectively) overnight in a 20 ml reaction volume, in presence of 5 mM MgCl$_2$. The reaction mixture was subjected to BN-PAGE analysis as described below. To test the role of Rx$^{NB}$-eGFP in NRC2 dimer activation in vitro, we expressed and purified both proteins separately from *nrc2/3/4* KO *N. benthamiana*. eGFP was included as a negative control for Rx$^{NB}$-eGFP. C-terminally FLAG-

tagged NbNRC2 and N-terminally FLAG-tagged Rx$^{NB}$-eGFP and eGFP were purified by FLAG affinity purification as described above, and 6 ml of 10 mM NbNRC2 were mixed in a final reaction volume of 20 ml with increasing amounts of Rx$^{NB}$-eGFP (1, 2, and 4 mM, respectively), in the presence of 5 mM ATP and 5 mM MgCl$_2$. Mixtures were incubated on ice overnight and then run on BN-PAGE as described below.

## BN-PAGE assays

For BN-PAGE, samples from in vitro studies described below were diluted as per the manufacturer's instructions by adding NativePAGE 5% G-250 sample additive, 4× Sample Buffer and water. After dilution, samples were loaded and run on Native PAGE 3% to 12% Bis-Tris gels alongside either NativeMark unstained protein standard (Invitrogen) or SERVA Native Marker (SERVA). The proteins were then transferred to polyvinylidene difluoride membranes using NuPAGE Transfer Buffer using a Trans-Blot Turbo Transfer System (Bio-Rad) as per the manufacturer's instructions. Proteins were fixed to the membranes by incubating with 8% acetic acid for 15 min, washed with water and left to dry. Membranes were subsequently reactivated with methanol to correctly visualise the unstained native protein marker. Membranes were immunoblotted as described above.

## Confocal microscopy

Three- to four-week-old plants were agroinfiltrated as described above with constructs of interest. The live cell confocal microscopy analyses were conducted 2 days after agroinfiltration. To image the infiltrated leaf tissue, they were excised using a size 4 cork borer, submerged in dH$_2$O, and live-mounted on glass slides. The imaging of the abaxial side of the leaf tissue was performed using a Leica STELLARIS 5 inverted confocal microscope equipped with a 63× water immersion objective lens. The laser excitations for GFP and mKOk tags were 488 nm and 514 nm, respectively. To prevent spectral mixing from different fluorescent tags when imaging samples with multiple tags, sequential scanning between lines was applied. All confocal images are single plane images.

## Bioinformatic and phylogenetic analyses

NLRtracker [12] output from 123 genome assemblies representing 38 species in 4 genera of the Solanaceae (nightshade) family [54,55] was imported in R, and 66,451 full-length NLR sequences were extracted; 23,872 NLR sequences with domain architectures of "CNL" and "CN" as determined by NLRtracker were retained. The NB-ARC domain (equivalent to NB domain and HD1-WHD domains in **Fig 1**) for these NLRs was also extracted from NLRtracker. Subsequently, the NB-ARC sequences of RefPlantNLR [12], tomato NRC helpers [56], and pepper NRC helpers [57] were added to the extracted NB-ARCs. The NB-ARC domain boundaries were retrieved from NLRtracker output and only sequences with NB-ARC sequences between 300 and 400 amino acids long were kept and truncated or unusually long NB-ARC sequences were discarded. This led to retaining 19,760 sequences. The resulting sequences were aligned using MAFFT v7.505 [--anysymbol] [58]. The alignment was then used to generate a phylogenetic tree using FastTree v2.1.11 [-lg] [59].

The NRC-network superclade containing 10,665 sequences (including helpers and sensors) was extracted from the CC-NLR phylogenetic tree by selecting the major branch containing both NRC reference helpers and sensors using Dendroscope v3.8.10 [Options > Advanced Options > Extract Subnetwork] [60]. The NRC superclade sequences were then realigned and used to create a new phylogenetic tree, from which the NRC helper clade was extracted using the same methods. The resulting 1,796 NRC helper sequences, without the reference

sequences, were deduplicated using CD-HIT v4.8.1 [-c 1.00] [61]. The deduplication was carried out to capture the highest diversity without introducing bias into the analysis due to over-represented sequences.

The resulting sequences were filtered based on length, retaining only sequences between 750 and 950 amino acids long. As a result, 993 sequences were kept. This decision was based on the length distribution within the NRC helper data and the length of the reference NRC sequences. The Geneious prime v2023.2.1 (https://www.geneious.com/) length graph feature was used for this purpose.

The NB-ARC domains from the remaining 993 sequences were combined with length filtered NB-ARCs (300 to 400 amino acids) of the reference NRC helpers from *N. benthamiana* (16 sequences), tomato (13 sequences), pepper (10 sequences), potato (42 sequences), and RefPlantNLR (14 sequences), resulting in 1,088 sequences in total, and aligned using MAFFT. The alignment was used to construct a new phylogenetic tree of deduplicated NRC helpers and reference sequences using FastTree. Finally, NRC helper subclades were extracted based on the presence of reference NRCs in well-supported major branches. Sequences that fell outside or between major clades were not included in the grouping of NRC helpers into subclades. The identified clades were then extracted using Dendroscope in the same way as before.

The full set of 1,092 NRC sequences consists of full-length sequences of the final set (993 sequences) and the unfiltered reference sequences of *N. benthamiana* (16 sequences), tomato (13 sequences), pepper (14 sequences), potato (42 sequences), and RefPlantNLR (14 sequences). These were also aligned using MAFFT [--localpair] to identify the dimer stretches in all NRC helpers. The alignment of the combined 1,092 sequences was trimmed to retain only important and well-aligned regions. This was accomplished using ClipKIT v2.0.1 [-m gappy] to remove sites with 90% gaps [62].

The dimerization stretches and conserved sequence motifs were mapped onto the alignment of all NRC helpers based on NbNRC2 and NLRtracker and extracted using the R scripts available at https://github.com/amiralito/NRC2Dimer [63]. Logo plots for stretches and sequence motifs were generated using the ggseqlogo R package [64].

## Shannon entropy analysis

Shannon Entropy values for all NRC helpers and each NRC clade separately were calculated using the Entropy R package [65] and visualised using the R scripts available at https://github.com/amiralito/NRC2Dimer [63].

## Identification of the unique residues in each NRC clade

To identify the unique residues in the NRC clades, we calculated the relative amino acid ratio among the NRC clades NRC2/3/4a/4b/4$_{other}$. We first divided the sequences into query and subject groups. Based on the trimmed alignment of NRC helpers, we calculated the amino acid frequency at each position in both query and subject groups. We then divided the amino acid frequency of the query group by that of the subject group at the same position. We only retained positions where one of the amino acid frequencies is more than 80% in the query group, and we regarded positions with a relative amino acid ratio of more than 50 as unique residues in the query clade. If an amino acid in the query group does not exist in the subject group, the ratio was output as infinite. The gaps were also ignored in this analysis. If all the query sequences had a gap at a position, the ratio was output as "NA." We iteratively calculated this relative amino acid ratio between an NRC clade and the other NRC clades, by alternating the query NRC clade. The scripts for this analysis are also available at https://github.com/amiralito/NRC2Dimer [63].

## Supporting information

**S1 Fig. NbNRC2 elutes as a dimer in analytical gel filtration assays.** Gel filtration assays with NbNRC2. C-terminally 6xHis-3xFLAG tagged NbNRC2 was expressed and purified from leaves of *nrc2/3/4* KO *N. benthamiana*. Purified protein was run on an S200 10/300 analytical column, and the estimated molecular weight of the NRC2 peak was calculated based on comparison of the elution volume to a protein standard curve ran on the same column. Elution volume (Ve) and molecular weight (Mw) in kDa are shown above the peak. A range of fractions below the peak, marked in green, were ran on SDS-PAGE and visualised using Coomassie Brilliant Blue staining (green panel). Approximate molecular weights (kDa) of the proteins are shown on the left of the green panel. The data underlying this figure can be found in **S6 Data**.
(EPS)

**S2 Fig. Cryo-EM image processing and 3D-reconstruction of NbNRC2 homodimers.** Schematic representation of cryo-EM data processing and 3D-reconstruction of resting state NRC2 homodimer. Detailed information on the data collection and processing as well as model building can be found in the pertinent Methods section and in **S1 Table**.
(EPS)

**S3 Fig. Representative map sections of NbNRC2 homodimer.** (**A**) Representative images taken from the map of the NbNRC2 homodimer showing the overall fit of the atomic model of NbNRC2 into the cryo-EM density map. (**B**) The extra density at NB domain which accounts for ADP is circled in red. (**C**) A small unaccounted globular density seen between WHD and LRR is shown in right bottom. This was modelled as IP6 in the recent SlNRC2 resting state structure [38]. (**D**) Closeup view of the post-processed map (left), showing helical grooves and bulk side chains at selected places along with the molecular model (right) fitted into it.
(EPS)

**S4 Fig. The helper NLR NbNRC2 forms a homodimer in planta.** (**A**) Overall molecular architecture and organisation of resting state NRC2 homodimer. Schematic of NbNRC2 domain architecture is shown above. Fit of NRC2 AlphaFold2 model into the cryo-EM density map is shown on the right. (**B**) Overall view of sharpened map by phenix_autosharp.
(EPS)

**S5 Fig. All NbNRC2 homodimerization interface mutants accumulate in planta.** NbNRC2, NbNRC3, and all NbNRC2 variants with mutations in the homodimerization interface were transiently expressed in *nrc2/3/4* KO *N. benthamiana*. Proteins were extracted 2 days post infiltration and run on SDS-PAGE assays and immunoblotted with the appropriate antisera labelled on the left. Approximate molecular weights (kDa) of the proteins are shown on the right.
(EPS)

**S6 Fig. NbNRC2^E243R homodimerization interface mutant triggers effector-independent autoactive cell death.** Quantitative analysis of cell death assays shown in **Fig 2A**, carried out in leaves of *nrc2/3/4* KO *N. benthamiana*. HR cell death was scored based on a modified 0–7 scale [25] at 5–7 days post agroinfiltration. NbNRC2 + Rx + CP was included as a positive control for cell death. HR scores are presented as dot plots, where the size of each dot is proportional to the number of samples with the same score (Count). Results are based on 3 biological replicates. Statistical analysis was carried out in R, using the BestHR package [66]. "a" or "b" on top of each treatment indicate statistically significant differences compared to the negative control NbNRC2. The data underlying this figure can be found in **S7 Data**.
(EPS)

**S7 Fig. NbNRC2 homodimerization interface mutants triggers effector-independent cell death when co-expressed with Rx.** Quantitative analysis of cell death assays shown in Fig 2A, carried out in leaves of *nrc2/3/4* KO *N. benthamiana*. Top panel shows cell death HR cell death was scored based on a modified 0–7 scale [25] at 5–7 days post agroinfiltration. Top panel features all NbNRC2 homodimerization interface mutants co-expressed with Rx. Top panel features all NbNRC2 homodimerization interface mutants co-expressed with Rx and with CP. In both cases, NbNRC2 + Rx and NbNRC2 + Rx + CP are included as negative and positive controls for cell death, respectively. HR scores are presented as dot plots, where the size of each dot is proportional to the number of samples with the same score (Count). Results are based on 3 biological replicates. Statistical analysis was carried out in R, using the BestHR package [66]. "a" on top of each treatment indicate statistically significant differences, compared to the negative control NbNRC2 + Rx. The data underlying this figure can be found in S8 Data. (EPS)

**S8 Fig. The nucleotide-binding domain of the sensor NLR Rx converts the NRC2 helper homodimer into a higher order oligomer in vitro.** (**A**) BN-PAGE assay with NRC2 incubated with ATP in vitro. Purified NbNRC2-FLAG (5 mM) was incubated overnight with no ATP or with increasing amounts of ATP (5, 10, or 15 mM). The experiment was repeated 3 times with similar results. (**B**) BN-PAGE assay with NRC2 incubated with $Rx^{NB}$ in vitro. Purified NbNRC2-FLAG (5 mM) was incubated overnight with increasing amounts of purified FLAG-$Rx^{NB}$-eGFP (0, 1, 2, and 4 mM, respectively). eGFP (4 mM) was included as a negative control. The experiment was performed 3 times with similar results. (**A, B**) In vitro reactions were run in parallel on native and denaturing PAGE and immunoblotted with the antisera labelled below each blot. Approximate molecular weights (in kDa) of the proteins are shown on the right. Red asterisk indicates signal at a molecular weight matching the previously reported NRC2 resistosome-like oligomer. (**C**) SDS-PAGE assays accompanying BN-PAGE assays. Affinity-purified proteins were run on SDS-PAGE assays and immunoblotted with the appropriate antisera labelled on the left. Approximate molecular weights (kDa) of the proteins are shown on the right. (EPS)

**S9 Fig. NRC sequences form a well-supported monophyletic clade within the CC-NLR and NRC superclade phylogenetic trees.** (**A**) NRC superclade including NRC helpers, Rx-type sensors, and SD-type sensors form a distinct, well-supported clade in a phylogenetic tree of 19,760 CC-NLRs from the Solanaceae family (S9 and S10 Data). (**B**) NRCs are monophyletic and part of the same well-supported clade inside the NRC superclade phylogenetic tree (S11 Data). (**C**) NRC sequences are divided into 15 distinct phylogenetic subclades based on the presence of reference NRC sequences (S12 and S13 Data). Numbers next to the major branches indicate bootstrap values. (EPS)

**S10 Fig. Sequence motif and Shannon entropy plots for 1,092 NRC sequences retrieved from 123 genome assemblies from 4 genera in the Solanaceae family.** Sequence motif and Shannon entropy plots for the MADA, P-loop, MHD motifs, and 8 dimerization stretches of 1,092 NRC sequences grouped into NRC clades based on phylogenetic relationships. The number of sequences and the distribution across the 4 genera within each clade are indicated on the right (S3 and S4 Data). (EPS)

**S11 Fig. Shannon entropy plots of full-length proteins calculated across all 1,092 NRC sequences and within each individual NRC clade.** The dimer stretches 1 and 2 are coloured by green and yellow, respectively. MADA, P-loop, and MHD motif are coloured by grey. Shannon entropy score of 1.5, indicated by the dotted line, was established as a threshold to distinguish between highly variable (>1.5 bits) and non-highly variable (<1.5 bits) amino acids, as established previously (31) (**S3** and **S4 Data**).
(EPS)

**S12 Fig. Contrasting mechanisms of activation between ZAR1 and the helper NLR NbNRC2.** The figure depicts the working model for activation of NRC2 homodimers. Unlike the functional singleton ZAR1, the helper NLR NRC2 exists as an autoinhibited homodimer in its resting form. NbNRC2 carries unique polymorphisms at the interface between its 2 protomers and exclusively associates with itself without hetero-dimerising with its NRC paralogs NbNRC3 or NbNRC4. Activation of upstream sensors and conditional exposure of their nucleotide-binding (NB) domains results in conversion of the NbNRC2 homodimer into a high molecular weight resistosome, presumably via homodimer dissociation into a putative primed monomer which subsequently oligomerizes. Like those observed for activated NRC4, the NRC2 oligomers may be hexameric in nature, which also differs from the pentameric complexes observed for activated ZAR1.
(EPS)

**S1 Table. Cryo-EM data collection, refinement, and validation statistics.**
(DOCX)

**S2 Table. Interface residues at 6 Å cut-off distance and their definition as amino acid stretches.**
(DOCX)

**S1 Movie. 3D view of NRC2 homodimer reconstructed from Cryo-EM micrographs (as separate file).** NB-ARC domain is shown in grey. LRR domain is shown in blue.
(MP4)

**S1 Data. Sequence alignment of NbNRC2, NbNRC3, NbNRC4, and NbZAR1 (as separate file).**
(FASTA)

**S2 Data. Genome metadata used in the phylogenetic analyses (as separate file).**
(XLSX)

**S3 Data. Curated 1,092 NRC sequences in FASTA format (as separate file).**
(FASTA)

**S4 Data. Entropy values for 1,092 NRC sequences and each NRC clade for the analysed dimerization stretches, sequences motifs, and the full-length proteins (as separate file).**
(XLSX)

**S5 Data. Frequencies of each amino acid within the NRC2, NRC3, NRC4other, NRC4a, and NRC4b clades compared to other NRC clades (as separate file).**
(XLSX)

**S6 Data. Numerical data underlying analytical gel filtration trace in S1 Fig.**
(XLS)

**S7 Data. Numerical data underlying cell death quantification in S6 Fig.**
(XLSX)

**S8 Data. Numerical data underlying cell death quantification in S7 Fig.**
(XLSX)

**S9 Data. List of CC-type NLR sequences and associated metadata (as separate file).**
(XLSX)

**S10 Data. Extracted NB-ARC (NBD and HD1-WHD) domain sequences of CC-type and reference NLRs (as separate file).**
(FASTA)

**S11 Data. NB-ARC (NBD and HD1-WHD) domain sequences of NRC superclade and reference NLRs (as separate file).**
(FASTA)

**S12 Data. NB-ARC (NBD and HD1-WHD) domain sequences of curated NRC sequences (as separate file).**
(FASTA)

**S13 Data. NB-ARC (NBD and HD1-WHD) domain sequences of reference NRC sequences from *S. lycopersicum*, *S. tuberosum*, *N. benthamiana*, *C. annuum*, and RefPlantNLR (as separate file).**
(FASTA)

**S1 Raw Images. Uncropped blots and gel scans underlying all figures in the manuscript.**
(PDF)

## Acknowledgments

We are very thankful to D. Lüdke (The Sainsbury Laboratory, Norwich, UK) for valuable discussions and critical reading of this manuscript. We thank C. McClune, (Stanford, California, USA) for useful discussions. We thank J. Richardson (John Innes Centre, Norwich, UK) for support and excellent maintenance of John Innes Centre Bioimaging facility. Negative stain EM data for this investigation were collected at JIC Bioimaging facility, which is supported by the Biotechnology and Biological Sciences Research Council (BB/CCG2240/1). We thank N. Lukoyanova (Birkbeck, University of London, London, UK) for support with Cryo-EM imaging. Cryo-EM data for this investigation were collected at ISMB EM facility, which is supported by the Wellcome Trust (202679/Z/16/Z and 206166/Z/17/Z). SK and MPC thank F. Sinatra and L. Messi for inspiration. We thank all members of the TSL Support Services for their invaluable assistance.

## Author Contributions

**Conceptualization:** Muniyandi Selvaraj, AmirAli Toghani, Chih-Hang Wu, Sophien Kamoun, Mauricio P. Contreras.

**Data curation:** Muniyandi Selvaraj, AmirAli Toghani, Hsuan Pai, Mauricio P. Contreras.

**Formal analysis:** Muniyandi Selvaraj, AmirAli Toghani, Mauricio P. Contreras.

**Funding acquisition:** Tolga O. Bozkurt, Sophien Kamoun.

**Investigation:** Muniyandi Selvaraj, AmirAli Toghani, Hsuan Pai, Yu Sugihara, Jiorgos Kourelis, Enoch Lok Him Yuen, Tarhan Ibrahim, Rongrong Xie, Abbas Maqbool,

Juan Carlos De la Concepcion, Lida Derevnina, David M. Lawson, Chih-Hang Wu, Mauricio P. Contreras.

**Methodology:** Muniyandi Selvaraj, AmirAli Toghani, Hsuan Pai, Yu Sugihara, Jiorgos Kourelis, Enoch Lok Him Yuen, He Zhao, Rongrong Xie, Abbas Maqbool, Juan Carlos De la Concepcion, Benjamin Petre, Mauricio P. Contreras.

**Project administration:** Muniyandi Selvaraj, Sophien Kamoun, Mauricio P. Contreras.

**Supervision:** Mark J. Banfield, Sophien Kamoun, Mauricio P. Contreras.

**Validation:** Muniyandi Selvaraj, Hsuan Pai, Mauricio P. Contreras.

**Visualization:** Muniyandi Selvaraj, AmirAli Toghani, Hsuan Pai, Enoch Lok Him Yuen, Mauricio P. Contreras.

**Writing – original draft:** Muniyandi Selvaraj, AmirAli Toghani, Yu Sugihara, Sophien Kamoun, Mauricio P. Contreras.

**Writing – review & editing:** Muniyandi Selvaraj, Hsuan Pai, Yu Sugihara, Jiorgos Kourelis, Enoch Lok Him Yuen, Tarhan Ibrahim, He Zhao, Abbas Maqbool, Juan Carlos De la Concepcion, Mark J. Banfield, Lida Derevnina, Benjamin Petre, David M. Lawson, Tolga O. Bozkurt, Chih-Hang Wu, Sophien Kamoun, Mauricio P. Contreras.

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
