## [Editor Report · Decision Letter 0]

7 Apr 2024

Dear Sophien, 

Thank you for submitting your manuscript entitled "Activation of plant immunity through conversion of a helper NLR homodimer into a resistosome" for consideration as a Short Reports by PLOS Biology.

Your manuscript has now been evaluated by the PLOS Biology editorial staff, as well as by an academic editor with relevant expertise, and I am writing to let you know that we would like to send your submission out for external peer review. 

I would actually ask you if you have several reviewer suggestions, since the Academic Editor did not have many other than the ones that have COI. We would like to move this as fast as possible given the other publications in revision.

Once your full submission is complete, your paper will undergo a series of checks in preparation for peer review. After your manuscript has passed the checks it will be sent out for review. To provide the metadata for your submission, please Login to Editorial Manager (https://www.editorialmanager.com/pbiology) within two working days, i.e. by Apr 09 2024 11:59PM.

Best wishes,

Melissa

Melissa Vazquez Hernandez, Ph.D.

Associate Editor

PLOS Biology

---

## [Decision Letter · Decision Letter 1]

7 May 2024

Dear Sophien,

Thank you for your patience while your manuscript "Activation of plant immunity through conversion of a helper NLR homodimer into a resistosome" was peer-reviewed at PLOS Biology. It has now been evaluated by the PLOS Biology editors, an Academic Editor with relevant expertise, and by three independent reviewers. 

In light of the reviews, which you will find at the end of this email, we would like to invite you to revise the work to thoroughly address the reviewers' reports. As you will see, the reviewers are mostly positive about the relevance and novelty of the study, yet they raise some concerns that would have to be addressed before we consider the manuscript for publication. Reviewers 1 and 2 would like you to validate the interfacial residues and the oligomerization. Specifically, Reviewer 2 has important concerns about the resolution of the cryo-EM structure and would like to see what the biological implications are in vitro or in vivo. Reviewer 3 thinks you should explore the role of ATP concentration and how Rx activates the oligomerization.

After discussing the reviews with the Academic Editor, we think that you should perform all the experiments suggested by the reviewers. Although in vivo biological implications could enhance the study, it would not be required for a Short Report and in vitro studies would be enough. Additionally, the cryo-EM resolution should be clarified and reported accurately, and we would also like you to consider an additional reference to measure ATP levels during ETI (10.1016/j.chom.2023.01.014).

Given the extent of revision needed, we cannot make a decision about publication until we have seen the revised manuscript and your response to the reviewers' comments. Your revised manuscript is likely to be sent for further evaluation by all or a subset of the reviewers.

**IMPORTANT - SUBMITTING YOUR REVISION**

*Re-submission Checklist*

*Published Peer Review*

*PLOS Data Policy*

*Blot and Gel Data Policy*

Sincerely,

Melissa

Melissa Vazquez Hernandez, Ph.D.

Associate Editor

PLOS Biology

REVIEWERS' COMMENTS:

Reviewer #1: 

In this manuscript (Short Reports), Selvaraj and co-workers report the cryoEM structure of the homodimeric NbNRC2, a helper NLR, at 3.9 A resolution. Incubation of NbNRC2 with high concentrations of ATP or the nucleotide-binding (NB) domain of the disease resistance protein Rx triggers NbNRC2 oligomerization. The authors show that the dimerization of the NbNRC2 is highly specific, as NbNRC2 does not pulldown other paralogs, such as NbNRC3, NbNRC4, or NbZAR1. This specificity arises from the distinct pattern of interfacial residues in individual NRC clades. Overall, this is an well done study that provides important insights into the structure and activation of NbNRC2. However, there are a few issues that should be addressed before publication.

1. The authors have discovered patterns of the interfacial residues that dictate the specificity of NbNRC2 and paralogs. Yet, no effort is made to validate this observation. Can the authors swap interfacial residues to alter the binding partner?

2. The discussion and experimental analysis of the Rx-triggered oligomerization of NbNRCs is confusing. Is it truly an oligomer or a hexamer? What is the experimental evidence? The native gel shift result in Figure 2 does not define the oligomeric state. Can the authors use crosslinking agents to define the oligomeric state?

Reviewer #2: 

Selvaraj and colleagues present their work determining the cryo-EM structure of NRC2 dimer purified from N. benthamiana as well as showing that three regions between NBD and LRR are involved in homo-dimerization. Given the genetic redundancy of NRC helper NLRs, the authors tested the hypothesis of Rx-mediated NRC2 resistosome formation using BN-PAGE on its own and confirmed that it is a likely scenario that is ATP-concentration dependent. The authors further demonstrated that NRC2 NLR prefers to form a homomeric complex in plants using Co-IP experiment even in the presence of the other NRC helper paralogs in its companion, and do not localize with the other paralog NLR4 in the plant cell when transiently expressed. The evolutionary analyses of the NRC members revealed that the residues at the dimeric interface underwent for diversification, implying that these residues might have played a role in insulating respective helper nodes from the others.

While the authors presented an interesting model of dimeric helper NRC2 as a major form existing in the resting state, which distinguishes itself from the first resistosome Zar1 model, the breadth of presented data appears to remain premature to strongly support the conclusion.

1. The authors reported the resolution as 3.9Å. However, there are several issues regarding the cryo-EM structure. The FSC curve plunges within a narrow resolution range indicating that the resolution may be overestimated, may be due to too tight mask or others. In addition, the maps shown in Supple Fig. 2 and Supple Fig. 3 do not align with a 3.9 Å resolution. These indicate that the resolution of the cryo-EM structure is highly overestimated. Furthermore, the authors refined the atomic model using the PHENEX real space refinement. As the real space refinement replies on the quality of the map, the current map quality would not allow to perform real space refinement. 

2. Despite the resolution issue in the cryo-EM structure, the structure delineates the interaction between two NRC2 molecules, which probably enables the authors to map the interaction residues in the dimerization surface. Therefore, it is surprising that the author did not attempt to validate the interface in-vitro and in-vivo. The author should design mutants to disrupt the interface and biochemically shows the mutants show a monomer. From the evolutionary analyses, the authors seem to already have identified relevant unique residues found in NRC2 and NRC3. It will be necessary to see at least if these evolutionary wired residues have impact on functional outcomes, e.g. strengthening dimer interfaces, helper function and specificity, when tested in each of the NRC molecules.

3. More importantly, biological implication of the dimeric interface shall be further corroborated. The authors speculated that multiple residues at the dimeric interface would play a role in insulating NRC2 signalling from the other helper NLRs action. Once the authors obtain which residues are critical in maintaining the dimeric interface of NRC2 homodimer (related to #3), the author should be able to show that how the stable dimeric formation contribute to fine-tune rather initially thought redundancy helper network or to facilitate NRC2-specific immune responses. Without in-vitro and in-vivo validation, this work is premature to be considered for a publication.

4. The authors shall consider revising what they consider as "resting" state as well as on the notion of the majority NRC2 form found in the plants being dimers. This can start with 1) clear indication of gel filtration data description with the strong notion of not detecting monomeric fractions; 2) indication of dimer configuration on the Figure 2 BN-PAGE gels, since the gels show two major forms (assuming the one around right below 480 marker should be dimer with the strong intensity, one can easily extrapolate that monomeric fraction might still exist as there is another band located below 242); 3) discussion on another possibility of NRC2-dimer being an intermediate/primed state towards the assembly of full resistosome. The schematic model on the dimer splitting to monomers to reconfigure to a hexamer (or higher-order oligomer) is based on NRC4 structure and largely on that of Zar1. It is not very convincing how much contribution of the observed dimer making to the assembly of full resistosome as compared to the monomer-constituting resting state model. If the monomeric mutation is feasible to obtain, the biological meaning of having the dimeric resting state (either promotive or inhibitory) will be self-explaining. 

Minor comment 

It is a convention not to capitalize C in cryo-EM.

Reviewer #3: 

This straightforward manuscript by Selvaraj et al. describes the solution of the cryo-EM structures of dimer and hexamer of helper NLR NRC2. Similar to the other solved CNL resistosomes, NRC2 forms a ring, but with different number of subunits (6 instead of 5). The manuscript is well written and the conclusions are nicely supported by the biochemical data provided. Here are some points for the authors to consider for either discussion or experimental enhancement of their story. 

1. In the model, the authors should include that with high ATP concentration, the NRC2 can form hexameric ring. Is this biological relevant? Have the authors tried to express NRC2 with high ATP concentration to see if HR reactions are changing in planta?

2. Does the NRC2 hexameric ring have calcium channel activity as with ZAR1? Is the pore size large enough? 

3. The NRC2-GFP imaging data is intriguing. Any sign of condensate formation or phase separation, which may allow high local ATP concentration? How about the clear nuclear localization? Is it biological relevant (are the proteins inside the nuclei inactive dimer or activated hexamer)? If it is the activated form, are they associated with nuclear envelop for activation?

4. It is unclear how the Rx NB activates the hexamer formation. Is this specific to certain Rx residues?

---

## [Decision Letter · Decision Letter 2]

12 Sep 2024

Dear Sophien,

Thank you for your patience while we considered your revised manuscript "Activation of plant immunity through conversion of a helper NLR homodimer into a resistosome" for publication as a Short Reports at PLOS Biology. This revised version of your manuscript has been evaluated by the PLOS Biology editors, the Academic Editor and the original reviewers.

Based on the reviews, we are likely to accept this manuscript for publication, provided you satisfactorily address the remaining points raised by the reviewers, specially that of reviewer #2 regarding the preprints. Please also make sure to address the following data and other policy-related requests.

a) The study is submitted as a Short Report which has a limit of 4 Figures. Currently your manuscript has 6 Figures. Please convert two of the main figures in supplementary figures, or combined with others so there are only 4 main Figures.

b) Please note that per journal policy, the model system/species (Nicotiana benthamiana) studied should be clearly stated in the abstract of your manuscript.

Please supply the numerical values either in the a supplementary file or as a permanent DOI’d deposition for the following figures:

Figure S1, S6, S7

d) Please cite the location of the data clearly in all relevant main and supplementary Figure legends, e.g. “The data underlying this Figure can be found in S1 Data” or “The data underlying this Figure can be found in https://doi.org/10.5281/zenodo.XXXXX”

e) We require the original, uncropped and minimally adjusted images supporting all blot and gel results reported in the Figures 1A, 2B, 3ABC, 4A, 5C, S1, S5, S8

We will require these files before a manuscript can be accepted so please prepare and upload them now. Please carefully read our guidelines for how to prepare and upload this data: https://journals.plos.org/plosbiology/s/figures#loc-blot-and-gel-reporting-requirements

f) Please ensure that your Data Statement in the submission system accurately describes where your data can be found and is in final format, as it will be published as written there.

g) Per journal policy, if you have generated any custom code during the course of this investigation, please make it available without restrictions upon publication. Please ensure that the code is sufficiently well documented and reusable, and that your Data Statement in the Editorial Manager submission system accurately describes where your code can be found.

We expect to receive your revised manuscript within two weeks. 

*Published Peer Review History*

*Press*

Sincerely,

Melissa

Melissa Vazquez Hernandez, Ph.D.

Associate Editor

PLOS Biology

REVIEWERS' COMMENTS:

Reviewer #1: 

The authors have adequately addressed previous critiques. The manuscript is now suitable for publication.

Reviewer #2: 

The revision was comprehensive in addressing all the issues we raised in the last round of revision. Mutation analysis did clarify functional relevance of keeping them as homodimers, which corroborated their discussion on functional diversification of the helper clade. In addition, this experiment also sensitized the helper to activate in the presence of the cognate sensor in the absence of the effector. As the authors discussed, this has much implication in optimizing disease resistance using NLRs. One last comment: Before making the manuscript public, I strongly suggest that the discussion points raised based on other bioRxiv preprints are checked for their validity, such as page 11 notions on proximity labeling data etc. 

Overall, this is a great work, highlighting the diverse ways of NLR activation, resting state of NLR configuration, as well as much in depth thoughts on NLR evolution and functional diversification. I thank the authors for their immense efforts to achieve this level of understanding. 

Reviewer #3: 

The authors did a good job addressing most of my concerns. Congratulations on a nice piece of work.

---

## [Editor Report · Decision Letter 3]

30 Sep 2024

Dear Sophien,

I hope you are doing great. Thank you for the submission of your revised Short Reports "Activation of plant immunity through conversion of a helper NLR homodimer into a resistosome" for publication in PLOS Biology. On behalf of my colleagues and the Academic Editor, Xinnian Dong, I am pleased to say that we can in principle accept your manuscript for publication, provided you address any remaining formatting and reporting issues. These will be detailed in an email you should receive within 2-3 business days from our colleagues in the journal operations team; no action is required from you until then. Please note that we will not be able to formally accept your manuscript and schedule it for publication until you have completed any requested changes.

PRESS

Sincerely, 

Melissa

Melissa Vazquez Hernandez, Ph.D., Ph.D.

Associate Editor

PLOS Biology
